# Sex and Body Colour Affect the Variation in Internal Body Temperature of *Oedaleus decorus asiaticus* in Natural Habitats in Inner Mongolia, China

Yumeng Cheng [1], Hongmei Li [1,2,*], Lulu Liu [1], Guangjun Wang [3], Haojing Gu [1,4] and Belinda Luke [5]

1   MARA-CABI Joint Laboratory for Bio-Safety, Institute of Plant Protection, Chinese Academy of Agricultural Science, Beijing 100193, China; chengym0222@163.com (Y.C.); lululiul@163.com (L.L.); graino@163.com (H.G.)
2   CABI East and Southeast Asia, Beijing 100081, China
3   State Key Laboratory for Biology of Plant Diseases and Insect Pests, Institute of Plant Protection, Chinese Academy of Agricultural Science, Beijing 100193, China; wangguangjun@caas.cn
4   The Administrative Office of Yuanmingyuan Park of Beijing Municipality, Beijing 100084, China
5   CABI, Egham TW20 9TY, UK; b.luke@cabi.org
*   Correspondence: h.li@cabi.org

**Abstract:** *Oedaleus decorus asiaticus* is one of the most harmful locusts in agricultural and pastoral areas in China. Plagues of this grasshopper can aggravate grassland degradation and cause huge damage to the livestock industry. Fungal biopesticides are seen as a suitable means of controlling grasshoppers and locusts. However, the efficiency of fungal biopesticides is dependent on temperature. Currently, there is limited knowledge on the thermal biology of this grasshopper in natural habitats. In this study, ground temperature measurements were made in conjunction with measurements of internal body temperatures using thermocouples and hand-held thermometers. The grasshoppers were randomly caught during the daytime in 2017 and 2018 in eight different locations in Inner Mongolia Autonomous Region, China. Our results indicated that the average internal body temperature of nymphs as well as adults of *O. d. asiaticus* was higher than the ground temperature and that it increases/decreases with increases/decreases in ground temperature, respectively, during the daytime. Moreover, the adult internal body temperature is significantly higher than that of the nymphs at different times of the day, specifically around 6:00, 10:00, 13:00, and 18:00. Female internal body temperatures were significantly higher than those of the males by an average of 0.90 °C. Additionally, the average internal body temperature of the brown morphs was higher than that of the green morphs by approximately 1.17 °C. These findings demonstrate that brown morph insects might be more tolerant of fungal biopesticides and hence the biopesticides may take longer to kill them. Hence, ecophysiological adaptations to climate change may affect how fungal biopesticides could be used in the future.

**Keywords:** grasshopper; internal body temperature; polymorphism; nymph; adult; mixed model; fungal biopesticides

## 1. Introduction

Locusts and grasshoppers are regarded as some of the most devastating pests affecting both agriculture and animal husbandry, and locusts can generally destroy green vegetation over millions of square kilometers within a short period [1–3]. In 2003, grasshoppers occurred in more than 26.66 million hm$^2$ of grassland in China, causing an annual direct economic loss of more than CNY 2.6 billion [4]. In recent years, plagues have intensified, and this may be due to climate change, increased anthropogenic activities, and improper development and utilization of natural resources. The grasshopper, *O. d. asiaticus*, has an extensive South Palearctic distribution, ranging from the Macaronesian Islands through the Mediterranean Basin to Central Asia [5]. It is particularly distributed in the Mongolian





Plateau and Transbaikal region of southern Russia [6]. *O. d. asiaticus* is regarded as the most dominant grasshopper species in the northern grasslands of China [7,8]. Currently, the control measures include chemical control, biological control, and ecological control. Biopesticides, based on the fungus *Metarhizium anisopliae*, have shown significant control of *O. d. asiaticus* [3]. However, the control efficiency of microbial pesticides is affected by environmental factors, such as temperature. Locusts have been shown to thermoregulate to control the development of the fungus [9]. Hence, measuring the internal body temperatures of orthoptera is important for understanding how quickly biopesticides are likely to work [10,11].

*O. d. asiaticus* is an ectotherm and hence may regulate its body temperature by basking in the sun or cooling off in the shade. Research has been carried out on body temperature constraints with respect to physiological processes [12–14], ecological factors [15–17], or morphological adaptations [18,19]. Numerous researchers have studied issues relating to body temperature in insects and other ectotherms, such as the thermal evolution of ectotherm body size [20–25], the effect of temperature on ectotherm ontogenetic growth and development [26,27], the physiological responses of ectotherms to daily temperature variation [28], ectotherm responses to climate change using thermal performance curves and body temperatures [29], the modelling of body temperatures of ectotherms [30], and body colour responses to body temperature [31–33]. Accordingly, the ability of ectotherms to flexibly alter physiological mechanisms in response to changes in environmental temperature (plasticity/acclimation/acclimatisation) determines their capacity to buffer performance and their fitness in conditions of environmental variation [34]. In its natural habitat, *O. d. asiaticus* has two different phenotypes: a gregarious type brown/dark brown in colour and a solitary type that is green in colour. It is crucial to understand the effects of environmental temperature on the body temperatures of the gregarious and solitary phases of the grasshoppers since temperature changes may directly influence their development, behaviour, and distribution and also the effectiveness of fungal biopesticides.

There is very limited knowledge about the body temperatures of the different phenotypes of *O. d. asiaticus*. The transformation from the solitary to the gregarious phase in response to a specific environment is the key factor leading to their flight and eruption [35]. Moreover, the gregarious grasshoppers may form visibly massive clouds and migrate long distances, causing unexpected losses [36,37]. Thus, in this study, we aimed to conduct the detection and observation of solitary and gregarious *O. d. asiaticus* in their natural habitats. For each non-migrated *O. d. asiaticus* individual body temperature, body colour and sex were measured. Furthermore, based on these original data, a biophysical model was developed to generalise the thermoregulatory response of *O. d. asiaticus*. The results of this study will provide basic information on the two phenotypes of *O. d. asiaticus*, which may help to predict its development and occurrence. In addition, understanding of internal body temperature may enable insights into how biopesticides will work under increased global warming.

## 2. Materials and Methods

### 2.1. Experimental Site

O. d. asiaticus inhabits the semiarid areas of the Xilingol League where the temperature fluctuations are extreme (high daytime temperatures >50 °C and relatively cool night-time temperatures <−40 °C; annual average temperature is 2.7 °C; average temperature in July is 21.0 °C). Eight typical locations were selected in the Xilingol League of Inner Mongolia Autonomous Region, China (Table 1), and the distance between two successive locations was at least 20 km. The annual precipitation is approximately 150–450 mm, most of which occurs between June and September, and the average temperature between May and September is 18 °C. The vegetation types in these areas are complex and diverse but mainly dominated by *Leymus chinensis*, *Stipa krylovii*, and *Stipa grandiflora*.

**Table 1.** Location of the study areas in the Xilingol League, Inner Mongolia, China.

| Experiment Locations | Longitude (E) | Latitude (N) | Altitude (m) |
|---|---|---|---|
| Scientific Observation and Experimental Station of Pests in Xilingol Rangeland, MARA | 116.001 | 43.950 | 984 ± 3 |
| The serpentine bay tourist area | 116.114 | 43.821 | 1066 ± 3 |
| Abag banner (40–50 km) | 115.380 | 43.877 | 1159 ± 3 |
| Shagouzi | 116.334 | 43.827 | 1167 ± 3 |
| Xilingol League reservoir | 116.089 | 43.848 | 1026 ± 3 |
| West Ujimqin banner 1 | 116.808 | 44.482 | 1023 ± 3 |
| Xilingol city | 116.284 | 44.057 | 1092 ± 3 |
| West Ujimqin banner 2 | 116.483 | 44.285 | 1040 ± 3 |

### 2.2. Measurements

The 'grab-and-stab' technique was used to detect the internal body temperature of *O. d. asiaticus* [31]. The brown and green nymphs or adults (Figure 1) were randomly captured by hand or with a sweep-net and their internal body temperatures ($T_b$) were immediately measured using a 0.125 mm diameter copper–constantan thermocouple connected to a hand-held, single-input, fast-response digital thermometer (Omega Engineering Ltd., Connecticut, American). At first, a small hole was made in the thorax of each insect with the tip of a 0.25 mm diameter hypodermic needle. Subsequently, the thermocouple was inserted to a depth of 2 mm, and the observation was recorded at the point where the temperature stabilized (almost immediately). The temperature of the microsite at the capture point was tested using thermocouples. The grasshoppers were released afterwards. Incidentally, the temperatures were recorded after 10–12 s of capture, and this time gap between the capture and the temperature recording may have affected the real body temperature. Moreover, along with the thorax temperature, we also noted the age, sex, body dimensions, and colour morph of each individual, as well as the time of day (hh: mm) when the measurement was recorded.

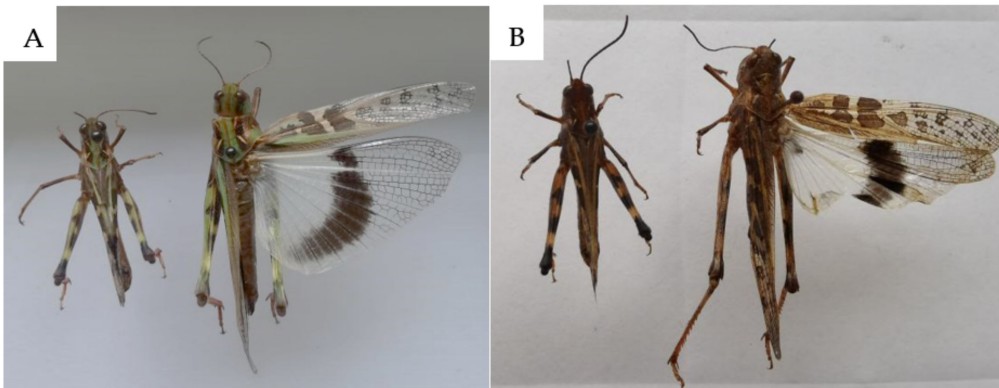

**Figure 1.** Images of green (**A**) and brown (**B**) *Oedaleus decorus asiaticus*. In each figure, the left specimen is a male and the one on the right is a female.

This study was carried out in July 2017 and again from June to August 2018. All sampling was performed on days without rain. The samplings were mostly performed between 6:00 and 20:00; however, we did not obtain any nymph data at 20:00.

### 2.3. Statistical Analyses

The time of day (hh: mm) recorded was transferred as hh during the data analysis. One-way analysis of variance (ANOVA) and Tukey's HSD post hoc test were used to assess nymph body temperature (nymph $T_b$), adult body temperature (adult $T_b$), and the temperature of the micro-site at different time points. Differences were considered

statistically significant at $p < 0.05$. The body temperatures of *O. d. asiaticus* were analysed using a generalised linear mixed-effects model. We fitted the daytime temperatures (centred at 12:00) as linear, quadratic, and cubic terms in the linear part of the model and fitted a random effect hour of recording [38]. Binary predictors were centred to zero, such that slopes could be used to estimate the differences among the groups, though the main effects remained interpretable in the presence of interactions [39]. Firstly, mixed model statistical analyses were performed to determine the influence of time, sex, and colour morph on the measured and calculated parameters. Furthermore, non-linear regressions in combination with an ANOVA were performed to assess and represent the dependence of these parameters on time, sex, and colour morph. A matched pair analysis was conducted by sex that was simple and efficient for controlling for temporal heterogeneity. We matched records for specimens of the same sex caught at the same location by the same team. These pairs, post hoc matched for variability in environmental conditions, were analyzed by paired *t*-tests. The average values for the evaluated parameters were derived from the fit curves. Post hoc pairwise differences between the fitted means were determined by the estimated 95% confidence intervals. Curve fitting and statistical analyses were performed using SPSS (19.0) statistical software and Prism.

## 3. Results

### 3.1. The Effect of Developmental Stage on the Internal Body Temperature of O. d. asiaticus

The datasets obtained from the same location in 2018 were used for this analysis. We observed that the average $T_b$ of both nymphs and adults of O. d. asiaticus varied greatly throughout the day in the field. Based on the analysis of the dataset developed from observation of 503 nymphs and 869 adults, internal body temperatures ranged from 13.4 to 44.9 °C for the nymphs and from 16.5 to 45.1 °C for the adults. The ground temperature varied from 10.3 to 57.1 °C. Incidentally, the $T_b$ of both nymphs and adults increased/decreased with increases/decreases in ground temperature, respectively, during the daytime (Figure 2).

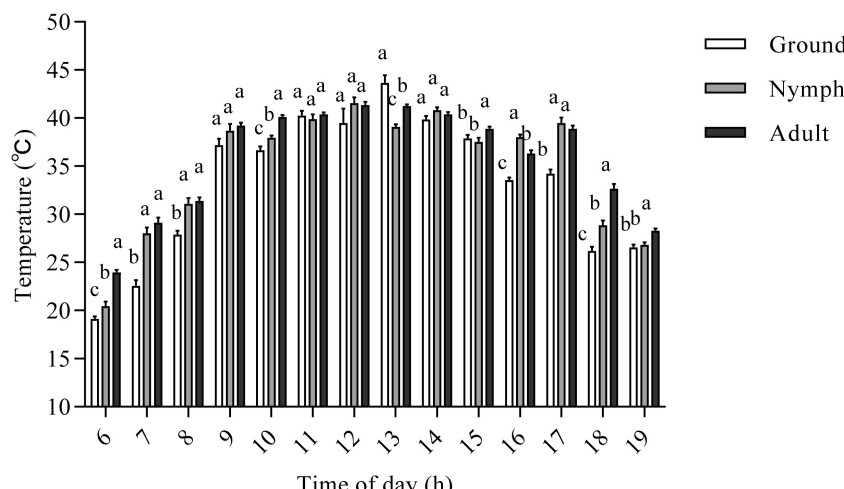

**Figure 2.** The average internal body temperature of *Oedaleus decorus asiaticus* in its natural environment and the variation in ground temperature as a function of the time of day (one-way analysis of variance (ANOVA) followed by Duncan's test, $p < 0.05$). Different letters indicate significant differences.

During the day, the average ground temperature was generally lower than the $T_b$ of O. d. asiaticus, and ground temperature varied significantly between 6:00 and 8:00, at 10:00, and between 16:00 and 18:00 ($p < 0.05$). Incidentally, the ground temperature was significantly higher than the $T_b$ of O. d. asiaticus at 13:00 ($p < 0.05$). Additionally, the $T_b$ of the nymphs was significantly higher than that of the adults at 16:00 ($p < 0.001$), while

the $T_b$ of the adults was significantly higher than that of the nymphs at 6:00 ($p < 0.001$), 10:00 ($p < 0.001$), 13:00 ($p < 0.001$), 15:00 ($p < 0.001$), 18:00 ($p < 0.001$), and 19:00 ($p < 0.001$). The $T_b$ of the adults was up to 3.8 °C higher than that of the nymphs, as observed at 18:00 (Figure 2).

### 3.2. Factors Affecting the Internal Body Temperature of O. d. asiaticus Adults

The body dimensions of 2408 O. d. asiaticus adults were measured in eight different regions in 2017 and 2018. Female average body lengths (3.14 ± 0.009 cm, range: 2.0 to 4.4 cm) were typically larger than male average body lengths (2.02 ± 0.004 cm, range: 1.2 to 2.6 cm).

Throughout the testing period, the females exhibited a higher average $T_b$ (17.5 °C to 45.3 °C) compared to the males (16.5 °C to 45.1 °C); moreover, both sexes indicated a normal distribution, with the highest $T_b$ recorded at 13:00 (Figure 3). The best-fit equation for the $T_b$ of the females was a second-order polynomial ($T_b = -9.01 + 7.65\,t - 0.29t^2$; $R^2 = 0.34$, $p < 0.001$; Figure 3) for both brown and green grasshoppers. Similarly, the $T_b$ of both green and brown males also followed a second-order polynomial equation ($T_b = -7.25 + 7.30\,t - 0.28t^2$; $R^2 = 0.44$, $p < 0.001$).

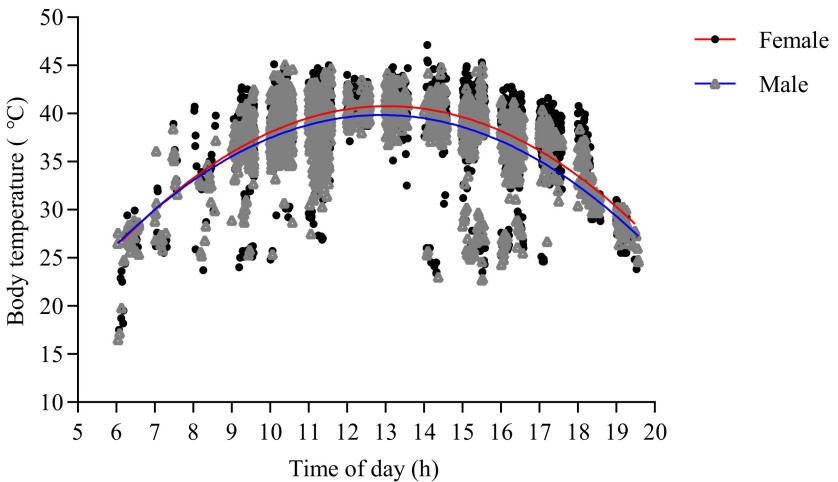

**Figure 3.** Daytime changes in the internal body temperatures of female and male *Oedaleus decorus asiaticus* specimens from eight different regions.

The average $T_b$ of brown female grasshoppers (22.9 °C to 47.1 °C) was always higher than that of the green female grasshoppers (17.5 °C to 44.3 °C), indicating that the brown grasshoppers gain more solar energy than the green ones during the daytime (Figure 4). Likewise, for the males, the average $T_b$ of brown grasshoppers (24.6 °C to 45.1 °C) was always higher than that of the green ones (16.5 °C to 44.8 °C), except for the temperatures recorded between 11:40 and 12:16 (Figure 4).

The daytime $T_b$ of brown and green grasshoppers can be divided into three phases, irrespective of whether they are female or male. The initial heat-up phase (6:00–10:00) was characterised by a steep increase in internal body temperatures at approximately 6:00; notably, the brown grasshoppers exhibited a significantly higher internal body temperature compared to the green individuals (females: F = 9.75, $p = 0.023$; males: F = 0.40, $p = 0.008$). In the next phase, the $T_b$ of the grasshoppers reached a static level (11:00–14:00) and the peak of the ectotherm body temperature (approximately 39.0–40.3 °C). During this phase, the grasshoppers did not increase their body temperatures any further; they maintained a relatively constant temperature, presumably through behavioural regulation to avoid overheating. Among females, brown individuals reached the static level 1 h earlier in the day (around 11:00) than green individuals (12:00, Figure 4), whereas among the males, both brown and green individuals maintained a relatively constant body temperature from approximately 11:00 to 13:00 (Figure 4). In the last phase, after 14:00, the $T_b$ of the

grasshoppers began to decrease. The difference in the $T_b$ variance between brown and green males was greater than the difference between the brown and green females. The $T_b$ of the male green grasshoppers declined quicker than that of the female green ones.

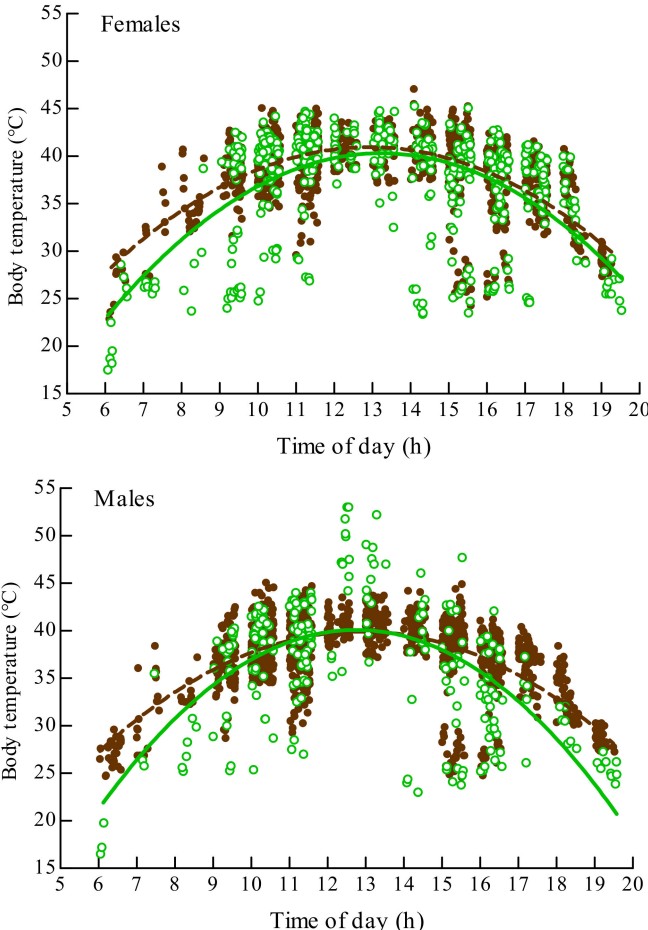

**Figure 4.** The 'time-course' of internal body temperature changes for females and males of *Oedaleus decorus asiaticus* from eight different regions. The green morphs are shown with open symbols; coloured lines show smoothed trends for brown (dashed lines) and green morphs (solid lines).

All the non-linear regressions for the body temperature of *O. d. asiaticus* as a function of time were significantly different (Table 2), indicating that internal body temperature is strongly influenced by the time of day.

**Table 2.** Non-linear body temperature model parameter estimates for *Oedaleus decorus asiaticus*.

| Sex | Morphs | Fit Curves | $R^2$ | df | F | *p* |
|:---:|:---:|:---:|:---:|:---:|:---:|:---:|
| Female | Brown | $T_b = -0.58 + 7.12\,t - 0.28\,t^2$ | 0.41 | 1122 | 361.93 | 0.00 |
| | Green | $T_b = -17.90 + 8.80\,t - 0.33\,t^2$ | 0.31 | 470 | 101.03 | 0.00 |
| Male | Brown | $T_b = -4.70 + 6.93\,t - 0.27\,t^2$ | 0.45 | 1281 | 545.00 | 0.00 |
| | Green | $T_b = -27.34 + 10.58\,t - 0.42\,t^2$ | 0.39 | 245 | 77.76 | 0.00 |

*3.3. The Effect of Morph on Internal Body Temperature of O. d. asiaticus Female and Male Adults*

Based on a linear mixed-model (LMM) analysis of $T_b$ datasets, the $T_b$ of the females was significantly higher than that of the males by approximately 0.90 °C ($t_{3092} = 3.09$, $p = 0.000$), with body colour having a significant influence on the $T_b$ of all grasshoppers; however, there is no statistically supported evidence for a sex-by-morph interaction

($\chi^2$ = 1.85, *p* = 0.78). In general, the $T_b$ of the green morphs was approximately 1.17 °C ($t_{1574}$ = 3.17, *p* = 0.002) cooler than that of the brown morphs.

The matched brown–green colour pair analyses revealed that the average $T_b$ of the green females was cooler than that of the brown females by 1.46 °C (411 matched pairs, *t* = 4.72, *p* < 0.001) and that the average $T_b$ of the green males was cooler than that of the brown males by 1.43 °C (235 matched pairs, *t* = 3.57, *p* < 0.001). Overall, the green morphs had a larger range of temperature fluctuations, dropping to approximately 32 °C, regardless of the sex of the grasshoppers (Figure 5).

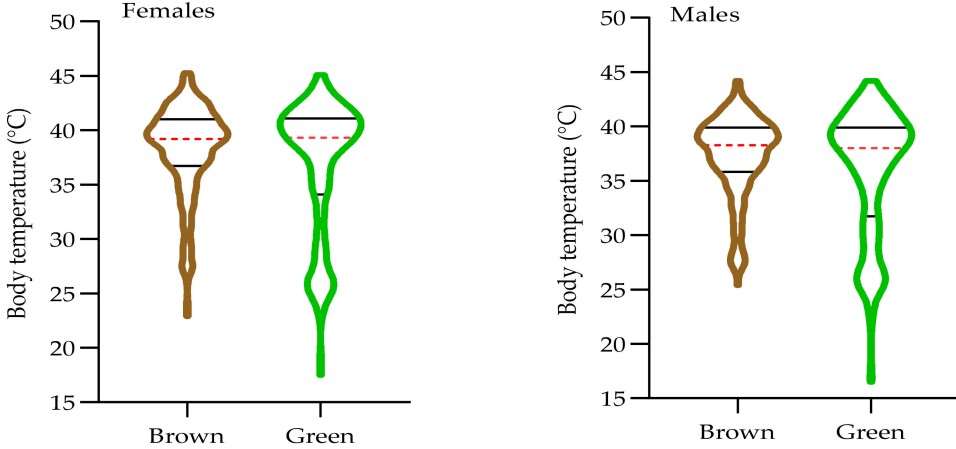

**Figure 5.** Violin plot of internal body temperature in different sexes of brown and green individuals of *Oedaleus decorus asiaticus*.

## 4. Discussion

This study shows that average $T_b$ for nymphs and adults of O. d. asiaticus was higher than the ground temperature, and that $T_b$ increases/decreases with increases/decreases in ground temperature, respectively, during the daytime. *Locusta migratoria manilensis* also displays a similar tendency for variation in $T_b$ in its natural habitat during the day [17]. During the investigation, it was observed that O. d. asiaticus quickly moves to warm, insolated patches of the soil surface to bask in the morning sun, where its body temperature is mainly affected by the ground temperature. Ectotherms rely on behavioural modifications to seek heat sources in the environment (e.g., sun-basking). *Oedaleus senegalensis* was shown to exhibit sunbathing behaviour, i.e., it tended to head east in the mornings to warm up [31], and Rskov et al. collected evidence of behavioural themoregulation in *L. migratoria* at dusk and dawn [39]. Moreover, this study revealed that the average $T_b$ of the nymphs depended more strongly on environmental conditions than did that of the adults. This may have been due to the nymphs mainly moving about on the ground and not being able to seek a high position on a plant to bask in the sun, while the adults can move to different levels of the canopy as well as in the air to seek out sunlight and hence warm up quicker. This trend was also seen during the daytime, where the $T_b$ of O. d. asiaticus adults was higher than that of the nymphs in most cases.

We observed that the $T_b$ of *O. d. asiaticus* females is higher than that of the males. This is probably related to the larger size of the females facilitating more heat absorption. Body size plays a fundamental role in defining the thermoregulatory challenges and limitations of organisms [21,22,40]. As compared to smaller individuals, larger individuals have a reduced surface-to-volume ratio, which helps them to retain their internal body temperatures for a longer period if challenged with a change in external temperature [23–25,41]. This concurs with the findings for alpine grasshoppers, the $T_b$ of the females having been found to be significantly higher than that of the males by an average value of 2.4 °C [38]. We also observed that the female brown grasshoppers reached a relatively stable body temperature earlier in the day than the female green grasshoppers. This suggests that colour-mediated

thermoregulation relates to green–brown polymorphism in orthopterans. The base temperatures at the beginning of the heat-up phase were different, with the green grasshoppers having a lower intial $T_b$. The green grasshoppers increased their body temperature at a speed similar to that of the brown grasshoppers, but during the sunset period the $T_b$ of green grasshoppers, especially in the males, decreased more rapidly than that of the brown grasshoppers. Therefore, the male as well as the female brown grasshoppers had stronger thermoregulation capabilities compared to the green ones. With increasing global warming, Zeuss et al. expect that dark-coloured insects may shift their distribution [42] and possibly retreat from certain areas [43], and/or even shift their habitat preference to shadier conditions on a small scale [44].

Internal body temperature can be different among different individuals of the same species due to differences in phenotype, body colour, and developmental stage. This study demonstrates that the $T_b$ of brown grasshoppers is higher than that of the green ones by approximately 1.4 °C. Köhler and Schielzeth found that the $T_b$ of brown morphs of *Gomphocerus sibiricus* and *Pseudochor thippus parallelus* was higher than that of green morphs by an average value of 1.5 °C [31]. Similarly, in the grasshopper *Kripa coelevriensis*, the blackish-brown form had a 4–5 °C higher temperature than the buff-coloured form when they were equally exposed to the sun [45]. In the chameleon grasshopper *Kosciuscola tristis*, a colour change from black to turquoise can produce a 0.23–0.55 °C difference in body temperature [32,33]. These results might thus, in principle, be explained by colour-specific differences in heat absorption, morph-specific differences in thermoregulatory behaviour (including microhabitat choice), or a combination of both. However, the biological significance of this difference is unclear, and more studies are necessary to directly test the hypotheses regarding the fitness benefits of colour-induced manipulation of body temperature. Recently, the implications of this colour variation have received particular attention with respect to determining evolutionary responses to climate change [46,47]. Zeuss et al. reported that the assembly of insect fauna in response to the thermal environment depends on the colour lightness of certain body parts, thereby demonstrating the importance of thermal energy in structuring insect assemblages, even across larger spatial scales [42]. This may help in predicting the effect of climate change [48–51], since insects can promote or hinder the evolution of colour change in response to climate change [40]. Our data show that brown individuals are slightly warmer on average and thus have larger activity windows under limited conditions (e.g., days without rain or days with variable conditions). It remains to be seen whether the difference is more or less pronounced under less favorable conditions.

Many previous studies have indicated that the performances of microbial agents and biopesticides are unstable due to a lack of information [52]. It is crucial to build a comprehensive understanding of the biology of hosts, pathogens, and their interactions. The potential implications for host–pathogen dynamics and the effectiveness of biocontrol methods involving pathogens are complex and may relate to host thermal behaviour, environmental factors, and the interactions between insect hosts and biopesticidal pathogens [9,53]. The harsh environment and the active thermoregulation of *Locustana pardalina* represent a severe challenge for pathogen growth and undoubtedly contribute to the variability in the performance of the LUBILOSA biopesticide [54]. Speed of kill is considered one of the major factors limiting the utility of mycoinsecticides in the control of many pests, including locusts and grasshoppers.

## 5. Conclusions

According to our findings, we determined that the $T_b$ of *O. d. asiaticus* adults is higher than that of nymphs in most cases, which suggests that adults can warm up quicker and take longer to cool down compared to nymphs. Additional analysis indicated that body size and brown–green colouration can affect the $T_b$ of *O. d. asiaticus*. Conclusively, this study established that female and brown morph insects might shift their habitat preference to more shady conditions under the impact of global warming. As global warming increase,

we might see more gregarious swarms in the northern grassland region of China. Therefore, it will be a good idea to apply biopesticide treatments against brown individuals earlier than treatments against green ones, as the former have the higher body temperature and may develop faster.

**Author Contributions:** Conceptualization, H.L.; methodology, H.L. and Y.C.; formal analysis, Y.C. and L.L.; data curation: L.L.; investigation, Y.C., G.W. and H.G.; writing—original draft preparation, H.L. and Y.C.; writing—review and editing, H.L., G.W. and B.L.; funding: H.L., G.W. and B.L. All authors have read and agreed to the published version of the manuscript.

**Funding:** This research was funded by grants from Newton UK–China Agricentres (Innovate UK/BBSRC; grant no.: 104906), the STFC (grant no.: ST/V000306/1), and China's Donation to the CABI Development Fund (grant no.: IVM10051).

**Institutional Review Board Statement:** Not applicable.

**Informed Consent Statement:** No applicable.

**Data Availability Statement:** Data can be provided upon request from the lead author.

**Conflicts of Interest:** The authors declare no conflict of interest.

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
