# Peer review of "Sex and Body Colour Affect the Variation in Internal Body Temperature of Oedaleus decorus asiaticus in Natural Habitats in Inner Mongolia, China"

_agriculture, doi:10.3390/agriculture12060878_

Round 1
Reviewer 1 Report
This is an interesting study on the variation in internal body temperature of Oedaleus decorus asiaticus in several natural habitats. This study has many applications including developing biological control methods which would consider species ecophysiological adaptations. Overall the manuscript is well-written and organized; all the figures and tables are well prepared; and all the images have high resolution. The authors provided the detailed introduction to the topic discussing the factors affecting the variation of grasshopper internal body temperature. The experiments are well designed, thoroughly conducted and well described.
I have only a few minor comments for the figures, which are easy to implement and would further strengthen the manuscript before publishing it:
Page 4. Figure 1. Please top align labels "A" and "B"
Figures 2 and 3. Please increase the font for the graph legends
Author Response
Dear Editor and Reviewer,
We much appreciate your favorite consideration and reviewers’ insight comments on our manuscript of “Sex and green-brown polymorphism affecting the variation in internal body temperature of Oedaleus decorus asiaticus in their natural habitats in Inner Mongolia, China” (Manuscript ID: agriculture-1717815). We have studied each comment and revised the paper carefully according to the editor’s and the reviewer’ comments. We hope this revision can make our paper more acceptable. The main corrections in the paper and responses to the editor’s and the reviewer’ comments are as following:
Reviewer #1:
This is an interesting study on the variation in internal body temperature of Oedaleus decorus asiaticus in several natural habitats. This study has many applications including developing biological control methods which would consider species ecophysiological adaptations. Overall the manuscript is well-written and organized; all the figures and tables are well prepared; and all the images have high resolution. The authors provided the detailed introduction to the topic discussing the factors affecting the variation of grasshopper internal body temperature. The experiments are well designed, thoroughly conducted and well described.
I have only a few minor comments for the figures, which are easy to implement and would further strengthen the manuscript before publishing it:
Page 4. Figure 1. Please top align labels "A" and "B"
Response: Thanks for the suggestions. Revised as suggested.
Figures 2 and 3. Please increase the font for the graph legends
Response: Thanks for the suggestions. The font for the graph legends increased to 9.

Reviewer 2 Report
1. Title needs to be changed to be more meaningful and related with research works
2. Importance of locusts (economic) shall be added as a line in abstract –scope or background of research.
3. Species name to be mentioned in full while mentioning for the 1st time (right from abstract)
4. A clear and concise methodology adopted to be included in abstract- lie how the authors determined the efficiency of pollinators.
5. Related sets of keywords to be used. Not one from the title
6.Introduction shall elaborate more on the locusts - causes, effects, decline rate plant production in % and suggested control measures if any.
7.Figures are presented in a good manner and informative
8. Tables could be formatted to be more understanding
9. Discussion - shall be rewritten - not proportionate with the number of results obtained - each category of results shall be explained, justified with appropriate citations.
Author Response
Dear Editor and Reviewer,
We much appreciate your favorite consideration and reviewers’ insight comments on our manuscript of “Sex and body colour affecting the variation in internal body temperature of Oedaleus decorus asiaticus under their natural habitats in Inner Mongolia, China” (Manuscript ID: agriculture-1717815). We have studied each comment and revised the paper carefully according to the editor’s and the reviewer’ comments. We hope this revision can make our paper more acceptable. The main corrections in the paper and responses to the editor’s and the reviewer’ comments are as following:
Reviewer #2:
Comments and Suggestions for Authors:
- Title needs to be changed to be more meaningful and related with research works
Response: Thanks for the suggestions. We revised as “Sex and body colour affecting the variation in internal body temperature of Oedaleus decorus asiaticus under natural habitats in Inner Mongolia, China” (Line 2-4, Page 1).
- Importance of locusts (economic) shall be added as a line in abstract –scope or background of research.
Response: Thanks for the suggestions. We added necessary information in the abstract (Line 19-20, Page 1).
- Species name to be mentioned in full while mentioning for the 1st time (right from abstract)
Response: Thanks for the suggestion. We changed them (Line 69, Page 2; Line 259, Page 8 ).
- A clear and concise methodology adopted to be included in abstract- lie how the authors determined the efficiency of pollinators.
Response: Thanks for the comments. The method information has been merged into the abstract (Line 21-25, Page 1).
- Related sets of keywords to be used. Not one from the title
Response: Thanks for the suggestion. The keywords have been changed (Line 40-41, Page 1).
- Introduction shall elaborate more on the locusts - causes, effects, decline rate plant production in % and suggested control measures if any.
Response: Thanks for the comments. We added that information in the introduction (Line 47-48, Page 2).
- Figures are presented in a good manner and informative
Response: Thanks for the comments. We fully agree with you.
- Tables could be formatted to be more understanding
Response: Thanks for the comments. We fully agree with you, and the geographic information of each site may help you all the readers to identify the field sites (Tables 1 & 2).
- Discussion - shall be rewritten - not proportionate with the number of results obtained - each category of results shall be explained, justified with appropriate citations.
Response: Thanks for the suggestions. We revised as suggested to make it clearly and more structurally following up the result part (Line 279-318, Page 9).
Reviewer 3 Report
The English needs significant editing.
The internal body temperature of a cold-blooded animal should be similar to that of its external environment. As someone who has never heard of someone measuring insect body temperature, I do not understand the merit of the research, as the results should be obvious. The project sounds uninteresting.
The authors need to provide an extensive literature review on all papers on the internal body temperature of Orthoptera and summarize this field to prove that this is a topic worth studying, and that the results are not expected to be “the insect’s body temperature matches that of the environment.” I would request no fewer than two whole paragraphs in the introduction and at least 20 new references on the subject of insect body temperature [unless the authors put this information in the discussion, in which case they should move it all up to the introduction].
A question I have about the conclusion [that dark insects will shift their distribution] is this: how does this work when individuals can change phase and color within their lifetime? Will dark morphs migrate more, or will phase changes appear less often in some parts of the world and more in others?
Ultimately the conclusion of the paper is that this species’s body temperature is influenced by that of the environment, which is an obvious conclusion for any insect. The authors need to do more to make this interesting.
Specific comments
37 - This statement is an exaggeration, especially as most grasshoppers are barely considered pests, and neither significantly affects animal husbandry as they are herbivores. It is best to say: “Locusts are serious pests that can destroy green vegetation over millions of square kilometers within a short period.”
43-45 Appears to be a mix of two sentences, and so is unreadable. Needs revision.
88 Table 1 is nice, but a figure showing these locations on a map might look good too.
Methods 2.2: How were the insects identified to species? Explain how you can be certain that they were this exact subspecies.
Also, how exactly was temperature at the microsite taken? Also with the thermocouple?
170 Delete “Furthermore,”
175-180 Dark colors absorb heat more than light, so it is not interesting at all that the dark locusts were warmer.
181 Delete “Incidentally”
280-281 I do not think insects can promote or hinder any form of evolution. Rather, say “color change may evolve in response to climate change”
Author Response
Dear Editor and Reviewer,
We much appreciate your favorite consideration and reviewers’ insight comments on our manuscript of “Factors affecting the variation in internal body temperature of Oedaleus decorus asiaticus in their natural habitats in Inner Mongolia, China” (Manuscript ID: agriculture-1717815). We have studied each comment and revised the paper carefully according to the editor’s and the reviewer’ comments. We hope this revision can make our paper more acceptable. The main corrections in the paper and responses to the editor’s and the reviewer’ comments are as following:
Reviewer #3:
The English needs significant editing.
Response: The manuscript has been edited by the company.
- The internal body temperature of a cold-blooded animal should be similar to that of its external environment. As someone who has never heard of someone measuring insect body temperature, I do not understand the merit of the research, as the results should be obvious. The project sounds uninteresting.
Response: Thanks for the question. Indeed, the body temperature of ectotherms is largely determined by environmental temperatures. However, we have to admit that behavioral thermoregulation may buffer many of the negative effects of extreme thermal conditions in parallel to plastic and evolutionary adjustments in physiological heat tolerance. Biopesticides control is one of the key methods to treat locusts and grasshoppers in China and abroad. The biopesticides such as Metarhizium anisopliae should play the high efficacy under suitable condition, and the pests as the host for the biopesticides. The host mainly avoids the invasion of spores and produces factors that are not conducive to invasion. This process is not only the interaction between pathogen and host, temperature is an important factor in the interaction relationship, and thermoregulation plays a very important role. Therefore, understanding the body temperature of Oedaleus decorus asiaticus in its natural state can provide a theoretical basis for understanding the interaction between biopesticides and Oedaleus decorus asiaticus.
- The authors need to provide an extensive literature review on all papers on the internal body temperature of Orthoptera and summarize this field to prove that this is a topic worth studying, and that the results are not expected to be “the insect’s body temperature matches that of the environment.” I would request no fewer than two whole paragraphs in the introduction and at least 20 new references on the subject of insect body temperature [unless the authors put this information in the discussion, in which case they should move it all up to the introduction].
Response: Thanks for the comment. We tried best to make this summary and merge into the introduction part. While we haven’t included all the references, as we are not sure for this point the journal would allow us to insert 20 literatures. The key and representative literatures have been cited and added in the reference part (Line 60-67, Page 2).
- A question I have about the conclusion [that dark insects will shift their distribution] is this: how does this work when individuals can change phase and color within their lifetime? Will dark morphs migrate more, or will phase changes appear less often in some parts of the world and more in others?
Response: Thanks for the suggestion. “The dark insects will shift their distribution” is the information from the citation and this is not the conclusion from our study, and our study is no related to distribution of insects. Zeuss et al (2014) used 473 European butterfly and dragonfly species, that dark-coloured insect species are favoured in cooler climates and light-coloured species in warmer climates. He further analysis 18 years distribution maps of dragonflies, then made the conclusion. We are planning to express that with the global warming, there is a potential distribution shift for the insects.
- Ultimately the conclusion of the paper is that this species’s body temperature is influenced by that of the environment, which is an obvious conclusion for any insect. The authors need to do more to make this interesting.
Response: Thanks for the point. Although insect body temperature is influenced by that of the environment, little research has been done on how it changes naturally and how insect conditions affect body temperature. According to our findings, we determined that the Tb of O. d. asiaticus adults is higher than that of the nymphs in most of the cases in their natural habitats. And we observed that the Tb of O. d. asiaticus females is higher than that of the males. Additional analysis indicated that the Tb of brown grasshoppers is higher than that of the green ones. The study of insect body temperature helps to understand the evolution of insect body color under global warming. Meanwhile, previously studies indicated that the body temperature is thought to be responsible largely for variable speeds of kill by mycoinsecticides in the field (Blanford et al., 1998, Lomer et al., 2001; Hunt, 2012). Thus this study is the kick-off work in this direction.
- 37 - This statement is an exaggeration, especially as most grasshoppers are barely considered pests, and neither significantly affects animal husbandry as they are herbivores. It is best to say: “Locusts are serious pests that can destroy green vegetation over millions of square kilometers within a short period.”
Response: Thanks for the suggestions. We would like to agree with your statement. From the global view, locusts have been well known as the causing heavily damages (e.g. desert locust, migratory locusts) comparing grasshoppers. According to the grassland situation in China, there are quite a few numbers of grasshoppers causing damages (e.g. Oedaleus decorus asiaticus). Now it is changed and modified as “ Locusts and grasshoppers are regarded as one of the most devastating pests affecting both agriculture and animal husbandry, and locusts can generally destroy green vegetation over millions of square kilometers within a short period”.
- 43-45 Appears to be a mix of two sentences, and so is unreadable. Needs revision.
Response: Thanks for the suggestion, we revised it (Line 52-56, Page 2).
- 88 Table 1 is nice, but a figure showing these locations on a map might look good too.
Response: Thanks for the suggestions. We agree that the map looks more visible comparing to table. As the study regions are not that famous, we thought the geographic information may provide precise information for everyone to mark on the map. Thus, we choose the table instead of map.
- Methods 2.2: How were the insects identified to species? Explain how you can be certain that they were this exact subspecies. Also, how exactly was temperature at the microsite taken? Also with the thermocouple?
Response: It is a good question. We use the morphometric characters to identify the species. Meanwhile, we also received the training by the local plant protection staff. The detection of body temperature was followed the below process. Once the grasshopper was caught, we first measured its body temperature and then double confirm the subspecies, then tested the temperature at the microsite with the thermocouple where the grasshopper was caught.
- 170 Delete “Furthermore,”
Response: Thanks for the suggestions, it was deleted.
- 175-180 Dark colors absorb heat more than light, so it is not interesting at all that the dark locusts were warmer.
Response: Thanks for the comments. In general, it is common knowledge about the brown/dark locusts were warmer. However, the body temperature between different colours of Oedaleus decorus asiaticus are not disclosed as well as for the females and males under natural conditions. Those information may help to set up the biopesticides application management.
- 181 Delete “Incidentally”
Response: Thanks for the suggestions, it was deleted.
- 280-281 I do not think insects can promote or hinder any form of evolution. Rather, say “color change may evolve in response to climate change”
Response: Thanks for the comments. We agree that Color change may evolve in reponse to climate change. Clusella-Trullas et al (2020) reviewed the existing evidence for contemporary adaptive evolution of insect color in response to climate change.
Clusella-Trullas, S.; Nielsen, M. The evolution of insect body coloration under changing climates. Curr. Opin. Insect Sci., 2020, 41, 25-32.
Round 2
Reviewer 2 Report
The manuscript became better after revision
Author Response
The english and spell has been revised by native speaker.
The introduction has been improved.
Reviewer 3 Report
The authors addressed many of my concerns in their response letter very well. I now understand the merit of this research: it's because temperature affects entomopathogenic fungi. However, they did not sufficiently repair the manuscript.
The new paragraph from line 79-86 contains many sentences on totally unrelated topics. There is a single line that says thermology is important to biopesticides. It does not have a single citation. That idea, that temperature affects pesticides like Metarhizium, is the single most important idea of the entire introduction: it is the main justification for this research. It deserves to be explained in an entire paragraph, not one, uncited sentence! Without a very detailed and heavily cited paragraph explaining to the reader the justifications for the study, the paper remains unjustified. The authors do not need to worry about reference number limits: the journal will allow more.
The next sentence says body temperature is a very good indicator for thermal biology. Since thermal biology is the study of body temperature, this is not necessary. The sentence also has nothing to do with the first sentence of the paragraph, which is about insect temperature and biopesticides. The next 3 sentences compare thermocouples and IR. Again, this has nothing to do with biopesticides, and also this paper does not mention IR ever again. This is not good writing. Every single sentence in a paragraph must be on the same subject.
In line 97-98, the authors mention body temperature’s effects on biopesticides again, briefly. Lines 104-107 also mention biopesticides. These ideas belong in the same paragraph as the first sentence of line 79-80, separate from the rest of this paragraph. The references also need to be explained: what do these references state? How and why does temperature affect biopesticides, in which insects? Read each paper and give short, one sentence summaries of how that paper relates to this manuscript.
While it requires time, I do suggest the authors delete the entire introduction and rewrite the whole thing, first by outlining the introduction so that every paragraph has one idea and one idea only. Then add sentences to each paragraph, with every paragraph having one idea and one idea only. The result will be clearer than the current introduction and worth the extra effort.
The discussion does not mention biopesticides at all. I really can’t stress this enough: the paper is very boring and uninteresting and unpublishable unless the authors can justify why the work is interesting. The paper does deserve to be published because temperature can affect entomopathogenic fungi, but the authors need to make this very, very clear. The authors cannot simply say “this information may help to set up biopesticides application management.” I do not believe this, so the authors must convince me. They need to state EXACTLY how biopesticides should be used now that we know that dark insects are warmer, and by how many degrees. Give information a farmer could use today, with very specific information on the perfect biopesticide application management system; or explain exactly what the next research project needs to be to produce the information a farmer could use tomorrow. Say as much as possible and give all the details that you can.
Finally, the English needs editing by a professional editing company or at least a native speaker.
Author Response
The authors addressed many of my concerns in their response letter very well. I now understand the merit of this research: it's because temperature affects entomopathogenic fungi. However, they did not sufficiently repair the manuscript.
The new paragraph from line 79-86 contains many sentences on totally unrelated topics. There is a single line that says thermology is important to biopesticides. It does not have a single citation. That idea, that temperature affects pesticides like Metarhizium, is the single most important idea of the entire introduction: it is the main justification for this research.
It deserves to be explained in an entire paragraph, not one, uncited sentence!
Without a very detailed and heavily cited paragraph explaining to the reader the justifications for the study, the paper remains unjustified.
The authors do not need to worry about reference number limits: the journal will allow more.
The next sentence says body temperature is a very good indicator for thermal biology. Since thermal biology is the study of body temperature, this is not necessary. The sentence also has nothing to do with the first sentence of the paragraph, which is about insect temperature and biopesticides. The next 3 sentences compare thermocouples and IR. Again, this has nothing to do with biopesticides, and also this paper does not mention IR ever again. This is not good writing. Every single sentence in a paragraph must be on the same subject.
In line 97-98, the authors mention body temperature’s effects on biopesticides again, briefly. Lines 104-107 also mention biopesticides. These ideas belong in the same paragraph as the first sentence of line 79-80, separate from the rest of this paragraph. The references also need to be explained: what do these references state? How and why does temperature affect biopesticides, in which insects? Read each paper and give short, one sentence summaries of how that paper relates to this manuscript. While it requires time, I do suggest the authors delete the entire introduction and rewrite the whole thing, first by outlining the introduction so that every paragraph has one idea and one idea only. Then add sentences to each paragraph, with every paragraph having one idea and one idea only. The result will be clearer than the current introduction and worth the extra effort.
Response: Thanks for the recommendation and suggestion. We accept your suggestion, and the introduction has been reorganized and rewritten. We remove the replications about the biopesticides topic in different paragraphs. The thermo biology of ectotherm has been summarized and include most of the literatures. This manuscript is focusing on the major grasshopper in northern grassland, China. This pest has two phenotypes, i.e. solitary (green colour) and gregarious (brown/ dark brown colour) types. The most of damages caused by brown individuals including the nymph and adult of grasshoppers. During outbreak periods, individual grasshoppers are attracted to each other, thereby forming aggregates or visibly massive clouds of gregarious insects, which can move in groups for long distances. Once the gregarious grasshoppers migrate from one place to another, the damages will be uncreditable. Thus, this study was conducted a baseline study to provide the preliminary information for sustainable control of grasshopper. However, there is very limited knowledge regarding the body temperatures of the different phenotypes of O. d. asiaticus. Therefore, in this study, we aimed to collect detailed field observations of solitary and gregarious locusts in their natural habitats. Furthermore, based on these original data, a biophysical model was developed to generalise the thermoregulatory response of O. d. asiaticus.
We hope the study results may provide the basic information for the prevention and control measures. For example, body temperature may play a significant role in the efficacy of intervening strategies, such as biopesticides, especially in the eradication of orthopteran (locust and grasshopper) pests, which can exhibit thermoregulation to elevate their body temperatures above the thermal range of a biopesticide.
The discussion does not mention biopesticides at all. I really can’t stress this enough: the paper is very boring and uninteresting and unpublishable unless the authors can justify why the work is interesting. The paper does deserve to be published because temperature can affect entomopathogenic fungi, but the authors need to make this very, very clear. The authors cannot simply say “this information may help to set up biopesticides application management.” I do not believe this, so the authors must convince me. They need to state EXACTLY how biopesticides should be used now that we know that dark insects are warmer, and by how many degrees.
Give information a farmer could use today, with very specific information on the perfect biopesticide application management system; or explain exactly what the next research project needs to be to produce the information a farmer could use tomorrow. Say as much as possible and give all the details that you can.
Response: Thanks for your suggestion. The paper is not the relationship between biopesticides and grasshopper, thus why there is not so much information about biopesticides. While, the we add some information how to use the body temperature information for the grasshopper treatment in the conclusion part--“Therefore, it will be good to apply the biopesticides treatment earlier against brown indi-viduals than the green ones as the former has the higher temperature and may develop faster. Meanwhile, it makes sense to control the grasshopper nymphs with good efficacy than the adult treatment with biopesticides.”
We added one paragraph about the effect of body temperature on growth of the biopesticide.
Finally, the English needs editing by a professional editing company or at least a native speaker.
Response: Thanks for your suggestion. The manuscript has been edited by the editing company!